# Myo-inositol nutritional supplement for prevention of gestational diabetes (EMmY): a randomised, placebo-controlled, double-blind pilot trial with nested qualitative study

Chiamaka Esther Amaefule [iD],[1] Zoe Drymoussi [iD],[1]
Francisco Jose Gonzalez Carreras,[1] Maria del Carmen Pardo Llorente,[2]
Doris Lanz,[1] Julie Dodds,[1] Lorna Sweeney [iD],[3] Elena Pizzo,[4] Amy Thomas,[1,5]
James Heighway,[1] Jahnavi Daru [iD],[1] Soha Sobhy,[1,5] Lucilla Poston,[6]
Asma Khalil,[7,8] Jenny Myers,[9] Angela Harden,[3] Graham Hitman,[10]
Khalid Saeed Khan [iD],[11] Javier Zamora,[12,13] Teresa Pérez [iD],[1,14]
Mohammed S B Huda,[15] Shakila Thangaratinam[13,16]

For numbered affiliations see end of article.

**Correspondence to**
Professor Shakila Thangaratinam;
s.thangaratinam.1@bham.ac.uk

## ABSTRACT

**Objectives** To determine the feasibility and acceptability of conducting a randomised trial on the effects of myo-inositol in preventing gestational diabetes in high-risk pregnant women.

**Design** A multicentre, double-blind, placebo-controlled, pilot randomised trial with nested qualitative evaluation.

**Setting** Five inner city UK National Health Service hospitals

**Participants** Multiethnic pregnant women at $12^{+0}$ and $15^{+6}$ weeks' gestation with risk factors for gestational diabetes.

**Interventions** 2 g of myo-inositol or placebo, both included 200 μg folic acid, twice daily until delivery.

**Primary outcome measures** Rates of recruitment, randomisation, adherence and follow-up.

**Secondary outcome measures** Glycaemic indices (including homoeostatic model assessment-insulin resistance HOMA-IR), gestational diabetes (diagnosed using oral glucose tolerance test at 28 weeks and by delivery), maternal, perinatal outcomes, acceptability of intervention and costs.

**Results** Of the 1326 women screened, 58% (773/1326) were potentially eligible, and 27% (205/773) were recruited. We randomised 97% (198/205) of all recruited women (99 each in intervention and placebo arms) and ascertained outcomes in 90% of women (178/198) by delivery. The mean adherence was 52% (SD 44) at 28 weeks' and 34% (SD 41) at 36 weeks' gestation. HOMA-IR and serum insulin levels were lower in the myo-inositol vs placebo arm (mean difference −0.6, 95% CI −1.2 to 0.0 and −2.69, 95% CI −5.26 to −0.18, respectively). The study procedures were acceptable to women and healthcare professionals. Women who perceived themselves at high risk of gestational diabetes were more likely to participate and adhere to the intervention. The powder form of myo-inositol and placebo, along with nausea in pregnancy were key barriers to adherence.

## Strengths and limitations of this study

⇒ Clinically relevant multicentre trial involving ethnically diverse high-risk women from inner city National Health Service hospitals in the UK, looking at myo-inositol versus placebo during pregnancy.

⇒ This pilot study, with a qualitative component, is based on a prospective protocol aimed at informing the feasibility of a full scale RCT (Randomised Controlled Trial) on the effect of myo-inositol versus placebo in preventing gestational diabetes in high-risk women.

⇒ Process outcomes relevant to recruitment, adherence and retention have been assessed in detail.

⇒ Self-reported adherence rather than objective methods (eg, pill-counting) was used.

⇒ The study allowed only limited evaluation of clinical outcomes due to the small sample size.

**Conclusions** A future trial on myo-inositol versus placebo to prevent gestational diabetes is feasible. The intervention will need to be delivered in a non-powder form to improve adherence. There is a signal for efficacy in reducing insulin resistance in pregnancy with myo-inositol.

**Trial registration number** ISRCTN48872100.

## INTRODUCTION

Primary prevention of gestational diabetes, a condition with high glucose levels first identified in pregnancy, is key to minimising maternal and perinatal complications associated with the condition.[1] Prevention of gestational diabetes can also reduce the risk of type 2 diabetes in mothers and their children in the long term.[2 3] Simple, easy to administer

and acceptable interventions are needed to prevent gestational diabetes.

Myo-inositol, a nutritional supplement that is present in fruits and fibre rich food and available as over-the-counter nutritional supplement, has the potential to prevent gestational diabetes through its insulin sensitising action. Few small trials have shown a promising role for myo-inositol with reductions in rates of gestational diabetes by up to 60%.[4 5] But, in addition to small sample sizes (<300 women), these trials were from a single country (Italy) and involved only Caucasian women.[5] The generalisability of these findings to high risk multiethnic populations who are most at risk of gestational diabetes is not known. A Cochrane review has called for further large trials on the effects of myo-inositol on gestational diabetes and complications in multiethnic populations. We also need to determine the costs and health service use associated with the intervention.[6]

Prior to undertaking a full-scale definitive trial on the effect of myo-inositol in preventing gestational diabetes, which requires substantial resources, we need to assess the feasibility of conducting such a trial beforehand. We undertook a pilot randomised trial comparing myo-inositol versus placebo, with a nested qualitative study, to assess the feasibility of the trial design, explore the acceptability and contextual issues around intervention delivery and trial procedures, and the potential effects on glycaemic, maternal, perinatal, cost and quality of life outcomes.

## METHODS

The EMmY trial was a multicentre, randomised, placebo-controlled, double-blind pilot trial with nested qualitative study and economic evaluation conducted with a prospective protocol and reported using Consolidated Standards of Reporting Trials guidelines.[7] The full details of the study protocol are published elsewhere.[8]

### Participants and setting

We screened pregnant women attending five inner city maternity units in London and Manchester. Women were eligible for recruitment if they were at least 16 years of age with a viable singleton pregnancy between $12^{+0}$ and $15^{+6}$ weeks' gestation, were able to provide written informed consent in English, and were considered to be at high risk of gestational diabetes as per the NICE (National Institute for health and Care Excellence) criteria.[9] We excluded women with type 2 diabetes, and those taking metformin or corticosteroids. Recruited women with a history of gestational diabetes underwent additional testing with OGTT (oral glucose tolerance test) or HbA1c, and were randomised after ruling out undiagnosed type 2 diabetes (OGTT fasting ≥5.6 mmol/L, 2-hour glucose ≥7.8 mol/L or HbA1c (Haemoglobin A1c) ≥48 mmol/mol).

### Intervention and control group allocation

The intervention group were provided with myo-inositol 2 g supplement with 200 µg folic acid, to be taken two times a day in a powder form mixed in water, approximately 1 hour before or after a meal. The placebo was an identical looking and tasting powder of Xylitol filler with 200 µg folic acid to be taken twice a day.

### Outcomes

The primary outcomes were the proportion of screened women who were eligible, the proportion of eligible women who were recruited and randomised, rates of adherence and follow-up. The secondary outcomes were acceptability of the intervention and trial to women and health professionals; laboratory outcomes related to glycaemia such as fasting and 2-hour plasma glucose levels in 75 g OGTT, homoeostatic model assessment-insulin resistance (HOMA-IR), serum insulin, adiponectin, leptin and urinary inositol at 28 weeks' gestation, and maternal and cord blood c-peptide at delivery. Other secondary outcomes included diagnosis of gestational diabetes at 28 weeks' gestation by modified IADPSG (International Association of Diabetes and Pregnancy Study Groups) and NICE criteria, maternal and perinatal morbidity and mortality (online supplemental appendix 1). The economic outcomes included costs (National Health Service (NHS) health and social care resources use) and quality of life. Other process outcomes included deviations from study protocol, completeness of data collection and the level of support required by the site team for trial delivery.

### Sample size

We expected that 1500 women would be booked for antenatal care each month at the participating hospitals, and at least 300 of those would be eligible. Assuming 1000 eligible women were approached, we expected 25% (250/1000) to be consented. We expected that 20% (50) of these would be women with a previous history of gestational diabetes, with abnormal HbA1C and/ or OGTT and would be excluded. This would result in 200 women randomised.

### Study conduct

We approached pregnant women at booking and screened them for eligibility. After obtaining written informed consent from eligible women, we collected baseline information on demographic and clinical characteristics, and women completed the validated European Quality of life 5-Dimensions 5-Level scale (EQ-5D-5L) questionnaire on quality of life measures. Women were randomised between $12^{+0}$ and $15^{+6}$ weeks' gestation using an online randomisation system (administered by epiGenesys, University of Sheffield) to either the myo-inositol or placebo treatment arm. We used a randomisation scheme based on permuted blocks of random block size (sizes 4, 6 and 8) stratified by participating site. No adaptive or minimisation strategies were used. Participants, healthcare

providers and researchers were blinded to the group allocation. We followed-up women in person or by phone at 20, 28 and 36 weeks' gestation and at delivery. All women were offered OGTT at 28 weeks' gestation to screen for gestational diabetes. Women self-reported adherence to the intervention using a paper-based diary or an optional mobile application. Where possible, we supplemented this information with a count of unused sachets. The trial was overseen monthly by a trial management group and biannually by the project steering committee (combining functions from a more traditional set up of a separate data monitoring committee and trial steering committee) reviewing trial progress and conduct. We prespecified the criteria for progression to a full-scale trial (online supplemental appendix 2).

## Analysis

Using an intention-to-treat approach, we summarised the feasibility outcomes using proportions with 95% CI and other descriptive statistics. For continuous laboratory outcomes, we calculated the effect sizes (eg, mean differences, MD) with 95% CI. We reported the rates of dichotomous clinical outcomes and costs in the two arms of the study. The impact of a previous history of gestational diabetes and the use of the mobile app on adherence were conducted as a post hoc exploratory analysis. All analyses were performed using R, V.3.6.1.

## Qualitative study

We explored the experiences and perspectives of women and healthcare professionals on the intervention and trial procedures, using observations and interview methods. Recruitment appointments were observed to ascertain women's specific motivations or concerns regarding participating in the trial, as well as contextual factors that may optimise or discourage participation. We invited eligible women who declined trial participation to give their reasons verbally to the researcher or complete an open-ended questionnaire for reasons for declining consent. We interviewed women and healthcare professionals involved in trial delivery to explore the barriers and facilitators of participation, adherence and retention in the study as well as trial delivery in interviews with health professionals. Interview data was transcribed and analysed using a thematic analytical approach.[10 11] Purposive sampling was used to ensure the women interviewed were from diverse age groups, ethnic backgrounds, clinical characteristics and levels of adherence (online supplemental appendix 3). We contacted women who withdrew from the trial, and midwives to discuss in depth the reasons for withdrawal.

## Economic evaluation

A cost–utility analysis of myo-inositol supplementation versus placebo 'within-trial' period was undertaken. The cost utility measures were the incremental cost per unit of change per quality-adjusted life-year (QALY) gained. Cost was assessed from the perspective of the NHS and

personal social services. The unit costs were retrieved from published sources, calculated in pound sterling based on 2018 costs. Resource use data was collected retrospectively. For each participant, their EQ-5D-5L health state was converted to a single summary index (utility value), constructing a utility profile based on the assumption of a straight-line relation between their utility values and each measurement point. For each participant, QALYs from baseline to 28 weeks were calculated as the area under the utility profile.

## Patient and public involvement

We sought input from Katie's Team, a women's health and childbirth patient and public involvement (PPI) advisory group,[12] who contributed to the study design and patient-facing documents for the pilot trial and the interview schedule for the qualitative study.[8] Patients and members of the public were not involved in the recruitment to and conduct of the study. We plan to disseminate results from the EMmY study to participants and members of Katie's Team, who can then circulate further through their networks and social media platforms.

## RESULTS

Over 6 months (February–September 2018), we screened 1326 women across all five participating sites, and randomised 198 women to either intervention or control arm (figure 1). The majority of participants in both arms were of ethnic minority origin (73% intervention; 60% control), over one-third were obese (37% intervention; 36% control), and around half (51% intervention; 40% control) were nulliparous. About 60% of all women had a first degree relative with type 2 diabetes, and one in ten (8% intervention; 11% control) had a history of previous gestational diabetes (table 1). Urinary inositol levels were higher in the myo-inositol than the placebo arm (MD 70.76 mg/L, 95% CI 11.2 to 130.8) (table 2).

## Primary outcomes

Of the 773 eligible women, 27% (205/773) consented to be involved in the study. Of these, 98% (198/205) were randomised to the trial after ruling out type 2 diabetes (figure 1). Figure 2 and online supplemental appendix 4 show the rate of recruitment and randomisation across all and individual participating sites respectively. The most common reason for women declining consent was a lack of interest in research (48%, 271/568) (figure 1). Outcomes were assessed in 75% (148/198) of women at 28 weeks, and 90% (178/198) of women were followed-up at delivery. The rates of adherence to the intervention in the myo-inositol (mean 53%, SD 45) and placebo (mean 50%, SD 43) groups were similar at 28 weeks. A sensitivity analysis showed that 75% (12/16) of women with gestational diabetes in previous pregnancies took more than 75% of myo-inositol sachets compared with 41% of women without a history of this condition (60/145). Six

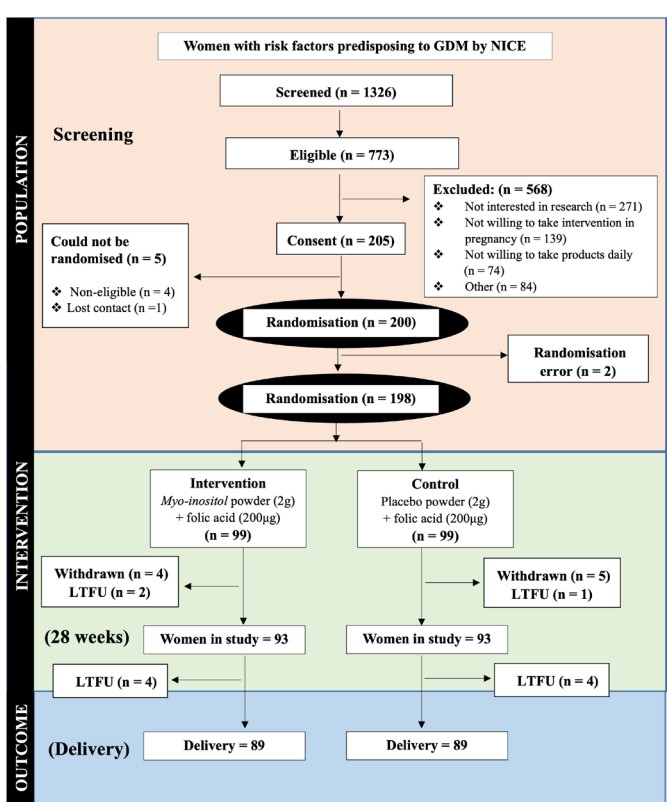

**Figure 1** CONSORT diagram for the EMmY study. CONSORT, Consolidated Standards of Reporting Trials; GDM, gestational diabetes mellitus; NICE, National Institute for Health and Care Excellence; LFTU, lost to follow up.

of the eight App users also took more than 75% of myo-inositol sachets.

## Secondary outcomes
### Glycaemic outcomes

We found significant reductions in HOMA-IR (MD −0.6; 95% CI −1.2 to 0.0) and levels of insulin (MD −2.69; 95% CI −5.26 to −0.18) in the intervention group versus control. There were no differences between the groups in other glycaemic estimates (table 2). Sensitivity analyses of women with over 50% and over 75% adherence to the intervention showed larger reductions in HOMA-IR and insulin levels (online supplemental appendix 5).

### Clinical outcomes

The proportion of women diagnosed with gestational diabetes were 14.1% (14/99) and 13.1% (13/99) in the intervention and control arms, respectively, at 28 weeks by modified IADPSG criteria. The overall diagnosis of gestational diabetes by delivery were 24.2% (24/99) and 20.2% (20/99) using any criteria (table 3). The rates of preterm birth in the myo-inositol and placebo groups were 6% (6/99) and 10.2% (10/99), respectively, and large-for-gestational age (LGA) were 6.1% (6/99) in intervention and 11.2% (11/99) in control arm. There were no reports of any maternal anaphylactic reaction or maternal death (table 3).

**Table 1** Baseline characteristics of EMmY participants randomised to myo-inositol versus placebo

| Baseline characteristics | Intervention (myo-inositol) N=99, n (%) or mean (SD) | Control (placebo) N=99, N (%) or mean (SD) |
|---|---|---|
| Demography | | |
| Age (years) | 31.4 (5.8) | 31.9 (5.6) |
| Gestational age at recruitment (weeks) | 12.4 (1.3) | 12.3 (1.5) |
| BMI (kg/m²) | 28.2 (6.4) | 27.9 (5.6) |
| Higher education | 73 (73.7) | 76 (76.8) |
| Ethnicity | | |
| White | 25 (25.3) | 40 (40.4) |
| Asian | 49 (49.5) | 48 (48.5) |
| Black | 16 (16.2) | 8 (8.1) |
| Mixed/others | 9 (9.1) | 3 (3) |
| Risk factors for GDM | | |
| GDM in previous pregnancy | 8 (8.1) | 11 (11.1) |
| Obesity (BMI ≥30 kg/m²) | 37 (37.4) | 36 (36.4) |
| Minority ethnic origin | 72 (72.7) | 59 (59.6) |
| Polycystic ovary syndrome | 9 (9.1) | 11 (11.1) |
| Previous macrosomic baby (>4.5 kg) | 4 (4) | 2 (2) |
| Family history of diabetes (first degree) | 60 (60.6) | 59 (59.6) |
| Current pregnancy | | |
| Parity | 0.8 (1) | 0.9 (0.9) |
| Nulliparous | 50 (50.5) | 40 (40.4) |
| Alcohol use | 1 (1) | 0 (0) |
| Tobacco use | 1 (1) | 3 (3) |
| Current use of supplements | 85 (85.9) | 89 (89.9) |
| Current use of medication | 24 (24.2) | 18 (18.2) |
| Aspirin | 8 (8.1) | 7 (7.1) |
| General clinical history | | |
| Pre-existing medical condition* | 4 (4) | 4 (4) |
| Family history of raised lipids (first degree) | 7 (7.1) | 8 (8.1) |

*Pre-existing medical conditions include autoimmune disease, blood/clotting disorder, cardiac problems, liver disease, lung disease, chronic hypertension and renal disease.
BMI, body mass index; GDM, gestational diabetes mellitus.

### Adverse outcomes

Vomiting was reported by 15.1% (15/99) of women in the intervention arm and by 9.1% (9/99) in the control arm. Other gastrointestinal symptoms were similar between

**Table 2** Glycaemia-related outcomes in the intervention and control arms of the EMmY trial

| Outcomes (unit) (intervention;control)* | Intervention (myo-inositol) Mean (SD) | Control (placebo) mean (SD) | MD (95% CI) |
|---|---|---|---|
| OGTT fasting (mmol/L) (81; 81) | 4.5 (0.5) | 4.6 (0.4) | −0.0 (−0.2 to 0.1) |
| OGTT 2 hours (mmol/L) (80; 81) | 6.1 (2) | 6.0 (1.3) | 0.0 (−0.5 to 0.6) |
| HOMA-IR (50; 62) | 2.13 (1.32) | 2.8 (2.05) | **−0.6 (−1.2 to 0.0)** |
| Insulin (mIU/L) (53; 63) | 10.63 (6.11) | 13.72 (8.89) | **−2.69 (−5.26 to −0.18**) |
| Leptin (µg/L) (53; 63) | 42.7 (21.6) | 38.3 (17.1) | 4.4 (−2.7 to 11.4) |
| Adiponectin (mg/L) (53; 63) | 6.6 (3.5) | 6.1 (3.8) | 0.5 (−0.9 to 1.8) |
| Urinary inositol (mg/L) (53; 65) | 265.7 (209.7) | 194.7 (118.7) | 70.76 (11.2 to 130.8) |
| c-peptide (µg/L) (53; 63) | 1538.5 (679.6) | 1804.3 (807.8) | −253.6 (−522.7 to 10.4) |
| c-peptide in cord blood (µg/L) (10; 6) | 457 (268.04) | 535 (296.02) | −101.74 (−358.17 to 202.17) |

*Number of women in the intervention and control group, respectively.
†Values in bold specifically described within secondary outcomes results.
HOMA-IR, homoeostatic model assessment-insulin resistance; MD, mean difference; OGTT, oral glucose tolerance test.

groups: diarrhoea was reported by 5.1% (5/99) and 4% (4/99) in the intervention and placebo arms respectively, and constipation by 5.1% (10/198) in both groups.

### Cost outcomes

Baseline EQ-5D-5L questionnaires were completed by all women randomised, and end of trial EQ-5D-5L questionnaires by 57% (56/99) in the intervention arm and 63% (62/99) in the control group. The total cost of trial (including the supplements, diagnostic tests, clinic visits and cost of birth/delivery) was very similar in both groups (£344 127 vs £344 814 in the intervention and control group respectively) and on average around £3500 per woman. The QALYs were similar in both groups with 0.53 QALYs over 28 weeks in the control and 0.51 QALYs in the intervention group. With no major difference in costs and QALYs between groups, an incremental cost-effectiveness ratio was not calculated and further sensitivity analysis was not explored. The costs of resources used by women in both groups as well as QALYs are reported in online supplemental appendix 6.

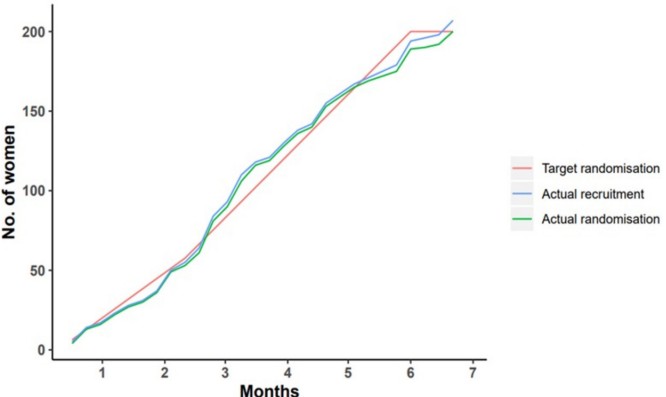

**Figure 2** Rate of recruitment and randomisation in all participating sites in the EMmY trial.

### Qualitative findings

We observed the recruitment of 28 participants across three sites. Nine women who did not want to participate in the study provided input on reasons for declining consent. Fifteen women consented to a qualitative research interview. Fourteen healthcare professionals were interviewed, including research midwives with responsibility for recruitment at each participating site, the academic trial coordinator, the research coordinator at participating sites and a clinical academic.

### Recruitment and participation

Women responded positively in the interviews about their experience of the recruitment process. They felt they received adequate information about the study to make a decision about participating, and that their questions and concerns were addressed. From our observations and interviews with women and research midwives, perceived risk of developing gestational diabetes was identified as a key factor influencing the decision about whether or not to participate in the trial. Women were more motivated to join the study if they believed they were at risk of developing gestational diabetes in their pregnancy, either due to a previous experience with the condition in an earlier pregnancy, or due to their awareness of gestational diabetes risk factors such as family history of diabetes or weight gain in pregnancy. Women without a history of gestational diabetes, or a family history of diabetes and those who felt they already engaged in healthy lifestyle practices did not typically view themselves to be at risk of developing gestational diabetes and were more likely to decline consent (even if the study had identified them as at-risk due to their ethnic background).

… in my first pregnancy I suffered a lot. I wasn't able to have anything…even if I had an apple…I used to climb the stairs up and down,…to…bring the sugar low…it's been a nightmare for me…I don't want that

**Table 3** Maternal and perinatal clinical outcomes in the EMmY study

| Outcomes | Intervention (myo-inositol) N=99, n (%) | Control (placebo) N=99, n (%) |
|---|---|---|
| **Maternal** | | |
| GDM (IADPSG) by 28 weeks | 14 (14.1) | 13 (13.1) |
| GDM (NICE) by 28 weeks | 14 (14.1) | 9 (9.1) |
| GDM (all definitions) by delivery | 24 (24.2) | 20 (20.2) |
| Pre-eclampsia | 3 (3) | 9 (9.1) |
| Hyperemesis | 3 (3) | 3 (3) |
| Third/fourth-degree tear | 2 (2) | 1 (1) |
| Postpartum haemorrhage | 17 (17.2) | 15 (15.2) |
| GA at delivery (weeks; mean SD) (88;89)* | 38.9 (3.2) | 38.8 (2.8) |
| Preterm delivery (<37 weeks) | 6 (6) | 10 (10.2) |
| Spontaneous vaginal delivery | 44 (44) | 45 (45.9) |
| **Perinatal** | | |
| Birth weight (g; mean SD) (87;88)* | 3260 (533.3) | 3251 (598) |
| Apgar score at 10 min (69;70)* | 10 (0.5) | 10 (0.8) |
| Respiratory distress syndrome | 1 (1) | 1 (1) |
| Large for GA | 6 (6.1) | 11 (11.2) |
| Small for GA | 18 (18.4) | 22 (22.4) |
| Admission to neonatal unit | 5 (5.1) | 3 (3) |

*Number of women in the intervention and control group, respectively.
GA, gestational age; GDM, gestational diabetes mellitus; IADPSG, International Association of Diabetes and Pregnancy Study Groups; NICE, National Institute for Health and Care Excellence.

to happen again…I'm just following it religiously, just to make sure… (Farihah, South Asian with previous gestational diabetes)

I've never had a patient with previous diabetes who declined the study. (Midwife A, Site 1)

The 'natural' make-up of myo-inositol as a vegetarian food supplement, rather than a drug, appeared to encourage trial participation. Many women felt reassured that it was safe to take the supplement in pregnancy and research midwives said it was ideal for the predominantly Muslim target population of the study, who only eat Halal foods.

… It was something natural…a natural ingredient, nothing too chemical or new or anything. So it didn't worry me. (Amber, White European with higher education level).

Nevertheless, some women cited 'fear of supplements' on the decliners' questionnaire as their reason for declining consent and some were concerned that participating in research involved testing the effect of new drugs, which could potentially harm the baby. Partners with these concerns sometimes discouraged participation even when the woman had indicated interest in being part of the research. High staff turnover rates in most participating sites affected study conduct. The midwives, research coordinator and trial coordinator suggested an increase in staff capacity or provision of support staff during more intense periods of study recruitment and follow-up. Research midwives highlighted that recruitment success was dependent on dissemination of trial information and engagement with the wider antenatal team.

## Adherence and retention

Several of the women interviewed experienced prolonged nausea and vomiting in pregnancy and discontinued the supplements when they felt these symptoms were made worse by drinking the supplement (several mouthfuls of 'sweet'-tasting, 'sandy'-textured liquid). Research midwives also cited pregnancy sickness as a major reason for participant non-adherence and withdrawal. A tablet form of the supplement was suggested by women and midwives as a preferable option in future studies. The twice-daily intake of the supplement and the time restrictions on taking it 1 hour before or after a meal were also major barriers to long-term adherence, as women struggled to embed the regimen into their busy daily routines.

…if it was in tablet form it would be easier for them to just swallow it, just like the Pregnacare… But because it's in liquid they have to take a mouthful and mouthful and mouthful… And the main reason for the withdrawal again is the fact that they were having very bad morning sickness. (Midwife G, Site 4)

Notably, adherence was reduced for some women after a normal or abnormal OGTT at the 28 weeks' visit. Those with a normal result felt the supplements were no longer needed and those with an abnormal result felt the strict regimen they had been following had not been effective.

This was less of an issue for women who felt most at risk of developing gestational diabetes.

We were not able to fully evaluate the use of the App, due to difficulties in the NHS Wi-Fi connectivity, which in many cases prevented the download of the app at the point of randomisation. The few women who downloaded the App felt the automated reminders were helpful with the regimen and critical to their adherence, while other participants without the app felt they would have adhered to the sachets more consistently if they had been supported with App reminders. They also felt they would have been more consistent in recording their adherence if they could have done so on their personal mobile phones via an App, rather than in a diary, which was likened to administrative work. By contrast, some women who were

not as comfortable with using technology felt the diary was easier and more straightforward.

> … I think an App or something would have been better… something that reminded me to take it… I just saw the booklet, it's just like an admin thing. That didn't help me. So an App might have done because I've got all the pregnancy Apps on my phone to remind me to take other things … (Amber, in full time employment)

## DISCUSSION
### Summary of key findings
Our multicentre pilot trial showed that it is feasible to recruit, randomise, adequately follow-up, and collect the relevant outcomes in a multiethnic inner-city pregnant population comprising of high-risk women, to evaluate the effects of myo-inositol on gestational diabetes. The trial procedures are acceptable to women and healthcare professionals. The powder form of the intervention was a barrier to adherence and women requested alternative forms such as a tablet. We observed a positive signal for efficacy in the myo-inositol arm than placebo for insulin resistance.

### Strengths and limitations
Our pilot randomised trial was based on a clear prospective protocol,[8] and we reported the findings using recommended guidelines.[7] We showed that it is feasible to recruit the required numbers of participants within planned time frames, and to cost, across multiple maternity units. More than half of our participants were from black and ethnic minority backgrounds, who are at most risk of developing gestational diabetes. We identified the barriers and facilitators to recruitment as reported by participating mothers and healthcare professionals. The trial procedures were robustly carried out, with good accuracy in data collection (online supplemental appendix 7). The outcomes were ascertained in over 80% of women in the control and intervention arms. The qualitative study identified the main factors to suboptimal adherence, which will need to be addressed in a full-scale trial. We showed that other outcomes such as clinical and cost-related outcomes can be collected to inform the full-scale trial.

Some of the limitations to the study included varying levels of recruitment rates across sites. Something to consider when planning the full-scale trial will be to increase staffing, to ensure that sites are fully supported during intense periods of participant recruitment and follow-up. There were more women of black and ethnic minority backgrounds in the intervention than control arms, who are more likely to have dysglycaemia in pregnancy. In a future full-scale trial, we plan to minimise by ethnicity to ensure both groups are fully balanced. Some of the qualitative insights suggested that adherence may have been improved if the App reminders had been made available for more women. Another limitation was not being able to assess the cord blood levels of the glycaemic markers in a larger proportion of participants.

### Interpretation of findings and comparison with existing literature
Our consent rates were consistent with other pregnancy trials involving multi-ethnic groups, citing similar reasons why eligible participants declined consent.[13 14] Previous trials on myo-inositol supplementation in pregnancy reported consent rates over 70% but had a predominantly Caucasian population.[15 16] Other pregnancy trials reported similar attrition rates to the EMmY study,[15 16] and made reference to phone-based reminders and diaries as crucial to adherence.[17 18] The EMmY study was not powered to detect any statistical differences in clinical or laboratory outcomes, including for gestational diabetes. Hence, we refrained from reporting comparative estimates for dichotomous outcomes. We identified the expected rates of gestational diabetes in the control arm that will inform the sample size calculations for the full-scale trial. We also observed that women continued to be diagnosed with gestational diabetes beyond 28 weeks, therefore a definitive trial would need to assess onset of gestational diabetes at any time in pregnancy along with critically important clinical outcomes as main outcomes.

### Clinical and research implications
Overall, the pilot study achieved 'green' status in rates of recruitment, attrition, and data collection, and 'amber' in adherence to the intervention at 28 weeks, as markers of progression to full-scale trial. Our qualitative study has identified the main reason for reduced adherence to be dissatisfaction with the powder form of the intervention. The findings were further discussed with the independent Steering Committee and Katie's Team, the PPi support arm of the study, who recommended use of myo-inositol tablets instead of powder when progressing to a full-scale trial.

We explored the differences in the continuous laboratory outcomes for signals of efficacy, as these would inform the choice of main outcomes for the full-scale trial. Despite lower than anticipated overall adherence to the intervention, the higher levels of urinary inositol in the intervention group makes the comparison of outcomes valid. In randomised trials on myo-inositol in pregnancy to prevent or treat gestational diabetes, myo-inositol showed a significant reduction in levels of fasting and postprandial blood glucose, HOMA-IR and insulinaemia.[19–21] We found a reduction in insulin and HOMA-IR in the intervention arm vs control, indicating a potentially increased insulin sensitivity with myo-inositol. Insulin levels and HOMA-IR appeared to reduce further with improved adherence. Our findings are in line with previous observations that myo-inositol reduces blood glucose levels by acting as an insulin sensitising agent.[16 21] We did not observe any differences in the fasting or 2-hour blood glucose levels, but improved adherence showed the

potential to reduce 2-hour glucose levels in the intervention group. It is likely that with improved adherence, any potential differences between groups will become clearer in any future studies.

Although there were no differences in the levels of c-peptide between the two arms, women who adhered to the intervention appeared to have a reduction in c-peptide levels, indicative of a reduction in insulin resistance. Participants in our study continued taking myo-inositol even after the diagnosis of gestational diabetes. We observed that fewer women in the intervention arm had LGA babies than in the control arm. The small sample size of this pilot trial means no statistical significance can be attributed to these estimates.[22] A future large scale trial will be needed to evaluate if the improvement in insulin sensitivity with myo-inositol contributed to the observed reduction in LGA babies.

## CONCLUSIONS

The EMmY pilot trial is the first UK based multi-centre trial involving ethnically diverse high-risk women from inner city NHS hospitals in a trial of myo-inositol versus placebo. We have reported on acceptability, cost and quality outcomes, as well as the potential effect of myo-inositol on gestational diabetes. There is an indication of efficacy of in lowering insulin resistance, with the potential for preventing gestational diabetes. In order to tackle the observed barriers to adherence, we plan to use a tablet form of myo-inositol in a large-scale definitive trial, with an internal pilot first demonstrating the intended levels of adherence.

We now need a large-scale definitive trial adequately powered to ascertain the effect of myo-inositol in a tablet form on gestational diabetes and adverse maternal and perinatal outcomes.

**Author affiliations**
[1]BARC (Barts Research Centre for Women's Health), Institute of Population Health Sciences, Queen Mary University of London Barts and The London School of Medicine and Dentistry, London, UK
[2]Department of Statistics and Operational Research, Complutense University of Madrid, Madrid, Spain
[3]Institute for Health and Human Development, University of East London, London, UK
[4]NIHR ARC (Applied Health Collaborations) for North Thames London, Department of Applied Health Research, University College London, London, UK
[5]Women's and Children's Health, Barts Health NHS Trust, The Royal London Hospital, London, UK
[6]Department of Women and Children's Health, Faculty of Life Sciences and Medicine, King's College London, London, UK
[7]Department of Obstetrics and Gynaecology, St George's Hospital, London, UK
[8]Vascular Biology Research Centre, Molecular and Clinical Sciences Research Institute, St George's University of London, London, UK
[9]Maternal and Fetal Health Research Centre, Manchester Academic Health Science Centre, The University of Manchester, Manchester, UK
[10]Centre for Genomics and Child Health, Blizard Institute, Queen Mary University of London, Barts and The London School of Medicine and Dentistry, London, UK
[11]Department of Preventive Medicine and Public Health, Universidad de Granada, Granada, Spain
[12]Clinical Biostatistics Unit and CIBER Epidemiology and Public Health, IRYCIS, Madrid, Spain
[13]WHO Collaborating Centre for Global Women's Health, Institute of Metabolism and Systems Research, University of Birmingham, Birmingham, UK
[14]Department of Statistics and Data Science, Complutense University of Madrid, Madrid, Spain
[15]Barts Health NHS Trust, The Royal London Hospital, Department of Diabetes and Metabolism, London, UK
[16]Birmingham Women's and Children's NHS Foundation Trust, Birmingham, UK

**Acknowledgements** The authors acknowledge Pharmasure for the donation of myo-inositol and placebo tablets to the EMmY study and Montuno software for the discounted Dosecast app. The authors are grateful to the patient advisers and site midwives for their contributions towards the study.The EMmY pilot trial was supported by the National Institute for Health Research (NIHR) Programme Grants for Applied Research. The views expressed are those of the authors and not necessarily those of the NHS, the NIHR or the Department of Health and Social Care. KSK is a Distinguished Investigator funded by the Beatriz Galindo (Senior Modality) Programme grant given to the University of Granada by the Ministry of Science, Innovation and Universities of the Spanish Government.

**Contributors** CEA: wrote the first and final version of the manuscript, collected and analysed the qualitative data. ZD, DL, JDo and JH: coordinated the study, contributed to the methodology and the final version of the manuscript. LS and AH: supervised the qualitative evaluation and has contributed to the final version of the manuscript. EP: developed and conducted the economic evaluation. EP also contributed to the final version of the manuscript. AT, JDa, SS, LP, AK, JM, GH, KSK and MSBH: provided clinical input and contributed to the final version of the manuscript. JZ, MdCPL and TP: developed and lead the statistical evaluation and contributed to the final version of the manuscript. FJGC: conducted the statistical analysis and contributed to the final version of the manuscript. ST: designed and developed the trial, reviewed the manuscript, has contributed to the final version of the manuscript, and is the guarantor of the manuscript.

**Funding** The EMmY trial is sponsored by Queen Mary University of London and funded by Barts Charity, grant number MGU0373. EP and AH are also supported by the NIHR Collaboration for Leadership in Applied Health Research at Barts Health NHS Foundation Trust (NIHR ARC North Thames). KSK is distinguished investigator funded by the Beatriz Galindo (senior modality) grant to the University of Granada by the Spanish Ministry of Education.

**Competing interests** None declared.

**Patient consent for publication** Not applicable.

**Ethics approval** This study involves human participants and was approved by London Queen Square Research Ethics Committee 17/LO/1741. Participants gave informed consent to participate in the study before taking part.

**Provenance and peer review** Not commissioned; externally peer reviewed.

**Data availability statement** Data are available on reasonable request. Release of data will be subject to a data use agreement between the contractor and the third party requesting the data. The data use agreement must detail the agreed use and appropriate management of the research data to be shared and include a requirement for recognition of the contribution of the researchers who generated the data and the original funder.

**ORCID iDs**

Chiamaka Esther Amaefule http://orcid.org/0000-0003-3864-2642

Zoe Drymoussi http://orcid.org/0000-0001-6683-9114

Lorna Sweeney http://orcid.org/0000-0002-1630-467X

Jahnavi Daru http://orcid.org/0000-0001-5816-2609

Khalid Saeed Khan http://orcid.org/0000-0001-5084-7312

Teresa Pérez http://orcid.org/0000-0003-0439-8952

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
