## [Reviewer comments · BMJ Open]

ARTICLE DETAILS

TITLE (PROVISIONAL)	Myo-inositol nutritional supplement for prevention of gestational diabetes (EMmY): a randomised, placebo-controlled, double-blind pilot trial with nested qualitative study
AUTHORS	Amaefule, Chiamaka; Drymoussi, Zoe; Gonzalez Carreras, Francisco; Pardo Llorente, Maria del Carmen; Lanz, Doris; Dodds, Julie; Sweeney, Lorna; Pizzo, Elena; Thomas, Amy; Heighway, James; Daru, Jahnavi; Sobhy, Soha; Poston, Lucilla; Khalil, Asma; Myers, Jenny; Harden, Angela; Hitman, Graham; Khan, Khalid; Zamora, Javier; Pérez, Teresa; Huda, Mohammed SB; Thangaratinam, Shakila

VERSION 1 – REVIEW

REVIEWER	Kulshrestha, Vidushi All India Institute of Medical Sciences, Obstetrics and Gynecology
REVIEW RETURNED	08-Apr-2021

GENERAL COMMENTS	Qualitative insights are novel. But otherwise literature has enough evidence regarding role of myoinositol in prevention of GDM.
---

REVIEWER	Yu, Yunxian Zhejiang University
REVIEW RETURNED	09-Apr-2021

GENERAL COMMENTS	This study uses the study design of RCT to examine the effect of Myo-inositol in preventing gestational diabetes mellitus (GDM) among pregnant women with high risk of GDM. The data shows HOMA-IR and serum insulin levels were lower in myo-inositol group than that in placebo group. Meanwhile, authors found the adherence of participants was affected, due to whether pregnant women perceived themselves at high risk of GDM and powder form of myo-inositol and placebo among the women with nausea. Although this pilot study has some limitations, so far, the effective intervention for preventing GDM was lack; wherever, most studies of GDM enrolled the general pregnant women, the women with high risk of GDM needed be more concerned. It is important that the current study focuses on this point. Hence this study indicates Myo-inositol is a potentially preventive method for GDM, and the study protocol is feasible in clinic. But the study with the large sample size needs to confirm this finding. Additionally, there are the following issues and suggestion to be concerned: 1. The intention-to-treat analysis must be conducted in this manuscript. Meanwhile, comparison of main outcomes between two groups had better adjust the potential confounders.
---

	2. It is unnecessary that the words that some persons (such as midwife, Amber) said were cited in RESULT section.
--	---

REVIEWER	D'Anna, Rosario Univ Messina
REVIEW RETURNED	22-Jul-2021

GENERAL COMMENTS	The Authors performed a randomized, double-blind, placebo controlled, multicentre pilot study on the use of myo-inositol for preventing GDM. First outcomes were the feasibility and the adherence to the study. Secondary outcomes were GDM rate in the 2 groups and other clinical outcomes. The Authors considered their study underpowered for the clinical outcomes, but there was no difference in GDM rate between groups thus I wonder how large should be the sample in the next study ? However, the study is interesting and the results encourages following studies Some points should be clarify:  1) line 142, In the myo-inositol group the supplement was taken without distinction before and after the meal. I disagree and I suggest to take the supplement only one hour before the meal 2) line 151, After glucose load with 75 gr, only 2 glucose values were assayed, lacking the 1-hour value, which is responsible of about 10-12% of GDM diagnosis. I strongly suggest to follow IADPSG criteria for OGTT. 3) In table 1 are reported baseline characteristics. It seems that the 2 groups are not comparable for the white ethnicity and for minority ethnic origin and this needs a statistical correction. 4) In table 2, Authors should provide HOMA and insulin levels evaluated at the beginning of the study In conclusion, future studies are needed because the data about insulin resistance and some obstetric outcomes such as the decrease rate of preterm birth and large for gestationale age fetuses in the treated group are promising.
--

VERSION 1 – AUTHOR RESPONSE

Reviewer #1

Qualitative insights are novel.

But otherwise literature has enough evidence regarding role of myoinositol in prevention of GDM.

We thank the reviewer for this comment. In addition to the role of myo-inositol in preventing GDM, our manuscript provides evidence on the feasibility of recruiting, retaining and assessing outcomes in a myo-inositol trial with multi-ethnic high-risk women within the NHS setting.

This evidence is lacking in the current literature, and as outlined by a Cochrane review are needed to investigate the effect of myo-inositol as preventative intervention for gestational diabetes. See “Page 5”, “Line 106-113”

Reviewer #2

This study uses the study design of RCT to examine the effect of myo-inositol in preventing gestational diabetes mellites (GDM) among pregnant women with high risk of GDM. The data shows HOMA-IR and serum insulin levels were lower in myo-inositol group than that in placebo group. Meanwhile, authors found the adherence of participants was affected, due to whether pregnant women perceived themselves at high risk of GDM and powder form of myo-inositol and placebo among the women with nausea. Although this pilot study has some limitations, so far, the effective intervention for preventing GDM was lack; wherever, most studies of GDM enrolled the general pregnant women, the women with high risk of GDM needed be more concerned. It is important that the current study focuses on this point. Hence this study indicates Myo-inositol is a potentially preventive method for GDM, and the study protocol is feasible in clinic. But the study with the large sample size needs to confirm this finding.

Additionally, there are the following issues and suggestion to be concerned:

1. The intention-to-treat analysis must be conducted in this manuscript. Meanwhile, comparison of main outcomes between two groups had better adjust the potential confounders.

The reviewer has highlighted that there was no effect observed with the pilot. However, as he/she mentions, our main aim is to pilot the study procedures and not assess the effectiveness of the intervention.

- 1) All participants randomised to the trial were included in the analysis, and in

2. It is unnecessary that the words that some persons (such as midwife, Amber) said were cited in RESULT section.

their allocation arm. Therefore, the intention-to treat analysis was used

Likewise, confounders will be adjusted for in the full-scale trial where it would yield more meaningful conclusions due to a larger sample size.

See "Page 8", "Line 187" for more clarity on the above.

- 2) We thank the reviewer for this comment. The name Amber is a pseudonym i.e not the real name of the participant who was interviewed or who gave that account. It is important to add verbatim data to the qualitative results to give the reader a sense of why an interpretation or conclusion was drawn from a particular viewpoint. It also constitutes a good standard for reporting qualitative data according to the Standards for Reporting Qualitative Research (SRQR) checklist

Reviewer #3

The Authors performed a randomized, double-blind, placebo controlled, multicentre pilot study on the use of myo-inositol for preventing GDM. First outcomes were the feasibility and the adherence to the study. Secondary outcomes were GDM rate in the 2 groups and other clinical outcomes. The Authors considered their study underpowered for the clinical outcomes, but there was no difference in GDM rate between groups thus I wonder how large should be the sample in the next study ? However, the study is interesting and the results encourages following studies Some points should be clarify:

We thank the reviewer for this comment. The sample size of the full-scale trial will be based on the event rate in the control arm observed in our study and the minimum clinically important difference that is relevant.

- 1) line 142, In the myo-inositol group the supplement was taken without distinction before and after the meal. I disagree and I suggest to take the supplement only one hour before the meal.
 - 2) line 151, After glucose load with 75 gr, only 2 glucose values were assayed, lacking the 1-hour value, which is responsible of about 10-12% of GDM diagnosis. I strongly suggest to follow IADPSG criteria for OGTT.
 - 3) In table 1 are reported baseline characteristics. It seems that the 2 groups are not comparable for the white ethnicity and for minority ethnic origin and this needs a statistical correction.
 - 4) In table 2, Authors should provide HOMA and insulin levels evaluated at the beginning of the study In conclusion, future studies are needed because the data about insulin resistance and some obstetric outcomes such as the decrease rate of preterm birth and large for gestational age fetuses in the treated group are promising.
- 1) We thank the reviewer for this suggestion. The prescription to administer the supplements away from food possibly one hour before or after a meal was given by the product manufacturers.
 - 2) We thank the reviewer for the suggestion. We have used the National Institute for Health and Care Excellence (NICE) criteria to diagnose GDM in our study which is the guideline that informs clinical practice in the NHS, UK, where our study is based. A pragmatic study design which includes the use of routine test and diagnostic criteria is essential for integration into routine practice. However, GDM outcomes will be reported based on the NICE and IADPSG criteria for comparative purposes as we have done in "Table 3", "Line 1 and 2" under "Maternal outcomes".
 - 3) We thank the reviewer for this observation. We are aware of the unequal distribution of baseline characteristics across arms, which is expected with small sample sizes. This issue would be minimised in the full-scale trial due to a larger sample size and with the use of the appropriate randomisation technique.
 - 4) We thank the reviewer for the suggestion and positive comment. We did not collect baseline data on HOMA-IR and insulin level, as our comparisons are between groups and not within group.

VERSION 2 – REVIEW

REVIEWER	Yu, Yunxian Zhejiang University
REVIEW RETURNED	18-Oct-2021

GENERAL COMMENTS	The unequal distribution of baseline characteristics across arms, which is expected with small sample sizes. Although this issue would be minimized in the full-scale trial, for this article, perhaps hierarchical analysis or multiple linear model can be supplemented.
--

REVIEWER	D'Anna, Rosario Univ Messina
REVIEW RETURNED	01-Nov-2021

GENERAL COMMENTS	Answers of Authors are satisfying. In my opinion no further changes of the text
---

VERSION 2 – AUTHOR RESPONSE

Reviewer #1

Qualitative insights are novel. But otherwise literature has enough evidence regarding role of myoinositol in prevention of GDM.

We thank the reviewer for this comment. In addition to the role of myo-inositol in preventing GDM, our manuscript provides evidence on the feasibility of recruiting, retaining and assessing outcomes in a myo-inositol trial with multi-ethnic high-risk women within the NHS setting. This evidence is lacking in the current literature, and as outlined by a Cochrane review are needed to investigate the effect of myo-inositol as preventative intervention for gestational diabetes. See “Page 5”, “Line 106-113”

Reviewer #2

This study uses the study design of RCT to examine the effect of myo-inositol in preventing gestational diabetes mellitus (GDM) among pregnant women with high risk of GDM. The data shows HOMA-IR and serum insulin levels were lower in myo-inositol group than that in placebo group. Meanwhile, authors found the adherence of participants was affected, due to

The reviewer has highlighted that there was no effect observed with the pilot. However, as he/she mentions, our main aim is to pilot the study procedures and not assess the effectiveness of the intervention.

whether pregnant women perceived themselves at high risk of GDM and powder form of myo-inositol and placebo among the women with nausea. Although this pilot study has some limitations, so far, the effective intervention for preventing GDM was lack; wherever, most studies of GDM enrolled the general pregnant women, the women with high risk of GDM needed be more concerned. It is important that the current study focuses on this point. Hence this study indicates Myo-inositol is a potentially preventive method for GDM, and the study protocol is feasible in clinic. But the study with the large sample size needs to confirm this finding.

Additionally, there are the following issues and suggestion to be concerned:

3. The intention-to-treat analysis must be conducted in this manuscript. Meanwhile, comparison of main outcomes between two groups had better adjust the potential confounders.

4. It is unnecessary that the words that some persons (such as midwife, Amber) said were cited in RESULT section.

3) All participants randomised to the trial were included in the analysis, and in their allocation arm. Therefore, the intention-to treat analysis was used

Likewise, confounders will be adjusted for in the full-scale trial where it would yield more meaningful conclusions due to a larger sample size.

See "Page 8", "Line 187" for more clarity on the above.

4) We thank the reviewer for this comment. The name Amber is a pseudonym i.e not the real name of the participant who was interviewed or who gave that account. It is important to add verbatim data to the qualitative results to give the reader a sense of why an interpretation or conclusion was drawn from a particular viewpoint. It also constitutes a good standard for reporting qualitative data according to the Standards for Reporting Qualitative Research (SRQR) checklist

Reviewer #3

The Authors performed a randomized, double-blind, placebo controlled, multicentre pilot study on the use of myo-inositol for preventing GDM. First outcomes were the feasibility and the adherence to the study. Secondary outcomes

We thank the reviewer for this comment. The sample size of the full-scale trial will be based on the event rate in the control arm observed in our study and the minimum clinically important difference that is relevant.

were GDM rate in the 2 groups and other clinical outcomes. The Authors considered their study underpowered for the clinical outcomes, but there was no difference in GDM rate between groups thus I wonder how large should be the sample in the next study ? However, the study is interesting and the results encourages following studies Some points should be clarify:

- 5) line 142, In the myo-inositol group the supplement was taken without distinction before and after the meal. I disagree and I suggest to take the supplement only one hour before the meal.
 - 6) line 151, After glucose load with 75 gr, only 2 glucose values were assayed, lacking the 1-hour value, which is responsible of about 10-12% of GDM diagnosis. I strongly suggest to follow IADPSG criteria for OGTT.
 - 7) In table 1 are reported baseline characteristics. It seems that the 2 groups are not comparable for the white ethnicity and for minority ethnic origin and this needs a statistical correction.
 - 8) In table 2, Authors should provide HOMA and insulin levels evaluated at the beginning of the study In conclusion, future studies are needed because the data about insulin resistance and some obstetric outcomes such as the decrease rate of preterm birth and large for gestational
- 5) We thank the reviewer for this suggestion. The prescription to administer the supplements away from food possibly one hour before or after a meal was given by the product manufacturers.
 - 6) We thank the reviewer for the suggestion. We have used the National Institute for Health and Care Excellence (NICE) criteria to diagnose GDM in our study which is the guideline that informs clinical practice in the NHS, UK, where our study is based. A pragmatic study design which includes the use of routine test and diagnostic criteria is essential for integration into routine practice. However, GDM outcomes will be reported based on the NICE and IADPSG criteria for comparative purposes as we have done in "Table 3", "Line 1 and 2" under "Maternal outcomes".
 - 7) We thank the reviewer for this observation. We are aware of the unequal distribution of baseline characteristics across arms, which is expected with small sample sizes. This issue would be minimised in the full-scale trial due to a larger sample size and with the use of the appropriate randomisation technique.
 - 8) We thank the reviewer for the suggestion and positive comment. We did not collect baseline data on HOMA-IR and insulin level, as our comparisons are between groups and not within group.

age fetuses in the treated group are promising.